# Mathematical Modeling for the Growth of *Salmonella* spp. and *Staphylococcus aureus* in Cake at Fluctuating Temperatures

Heeyoung Lee [1] , Jin Hwa Park [1] , Yu Kyoung Park [1] and Hyun Jung Kim [1,2,*]

1    Korea Food Research Institute, 245 Nongsaengmyeong-ro, Wanju 55365, Korea; hylee06@kfri.re.kr (H.L.); parkjinhwa@kfri.re.kr (J.H.P.); dbrud7922@naver.com (Y.K.P.)
2    Department of Food Biotechnology, University of Science and Technology, 217 Gajeong-ro, Daejeon 34113, Korea
*    Correspondence: hjkim@kfri.re.kr

**Abstract:** This study aimed to develop dynamic mathematical models to predict the growth of *Salmonella* spp. and *Staphylococcus aureus* in a cake under fluctuating temperatures. Among the nine different types of cakes frequently served during school meals, one type of cake was selected based on bacterial growth and water activity. Cocktails of *Salmonella* spp. and *S. aureus* were inoculated in the samples and stored at 4–35 °C for up to 336 h. The growth of *Salmonella* spp. and *S. aureus* was observed above 20 and 15 °C, respectively. The bacterial cell counts were fitted in the Baranyi model, and the maximum specific growth rate ($\mu_{max}$; log CFU/g/h) and lag phase duration (LPD; h) were analyzed using a polynomial model as a function of temperature ($R^2$ = 0.968–0.988), and the performance of the developed models was appropriate. Furthermore, dynamic models were developed, and the predictions were acceptable in changing the temperature, indicating that the developed dynamic models can successfully predict the outcomes of *Salmonella* spp. and *S. aureus* in cake. These results provide useful information for assessing and managing microbial risk in foods by predicting the behavior of *Salmonella* spp. and *S. aureus* in cake, especially in changing temperature.

**Keywords:** *Salmonella* spp.; *Staphylococcus aureus*; mathematical model; cake; dynamic temperature





## 1. Introduction

Refrigerated baked desserts, such as cake, are considered ready-to-eat (RTE) as they do not need any additional preparation prior to consumption. These desserts are popular for their convenience; unfortunately, some of them are associated with outbreaks of foodborne illnesses in the USA, Canada, and other countries worldwide [1–3]. A massive *Salmonella* outbreak, including 2207 cases, resulted from the consumption of contaminated chocolate cake in Korea in 2018, suggesting the requirement of risk management for the cakes. A previous study reported that egg white, a chocolate cake ingredient, was contaminated with *Salmonella* Thomson and was the cause of the outbreak [3]. Another outbreak of foodborne infection resulted from the consumption of contaminated cream cake, supporting the need for developing risk management options for cakes and other bakery products. The foodborne outbreaks resulted in 216 gastroenteritis cases in Singapore in 2007. *Salmonella enterica* subspecies enterica serotype Enteritidis was identified as the causative agent, and the vehicle of transmission was revealed to be cream cakes made in a bakery and sold at its retail outlets [4]. Centers for Disease Control and Prevention (CDC) reported that these infections from five states resulted from the same strain of *Salmonella* Agbeni as the *Salmonella* strain identified in the cake mix [2,5]. The New South Wales Food Authority, Australia, evaluated the microbial quality of bakery products including cakes because bakery products are frequently consumed but associated with foodborne outbreaks (NSW Food Authority, 2007). They reported that 2.2% of 696 samples were not acceptable according to the guidelines by the Food Standards Australia New Zealand [6]. Compared to many *Salmonella* foodborne outbreaks, occurrences of *S. aureus* infection are relatively

low, but there have been reported cases in bakery products: twenty-four people infected with *S. aureus* due to Chantilly cream dessert in Italy [7], and *S. aureus* detected in cake made in a Chicago-area cake bakery in 2011 [8]. *S. aureus* is easily found in the environment; thus, *S. aureus* foodborne illness have been occurring constantly. However, the bacteria outbreaks have been decreased from 372 (2010) to 52, ranked the sixth in Korea [9].

*Salmonella* are a common cause of foodborne illness worldwide. Nearly one in three foodborne outbreaks in the EU in 2018 was caused by *Salmonella* spp. [10], and the Center for Disease Control and Prevention (CDC) has estimated that *Salmonella* causes 1 million foodborne illnesses every year in the United States [11]. Furthermore, *Staphylococcus aureus* is a major cause of foodborne infections, releasing toxins in contaminated food items. *S. aureus* infections have been reported in the skin or nose in 25% of individuals, thus suggesting potential cross-contamination of food items by infected individuals [12]. Therefore, *Salmonella* and *S. aureus* contamination may have occurred in the cake considering the present outbreak, and bacterial growth may occur upon temperature abuse, resulting in foodborne diseases. Mathematical models have been used to describe the kinetic behavior of pathogenic bacteria under different conditions of temperature, pH, and water activity ($a_w$) [13]. The kinetic model, a mathematical model, can estimate kinetic parameters, the maximum specific growth rate ($\mu_{max}$), and lag phase duration (LPD), and these kinetic parameters can be used for predicting bacterial growth in food during storage [14]. Predictive modeling for the behavior of foodborne pathogens may provide useful information for assessing and managing microbial risk in foods [15]. Sufficiently accurate predictions of bacterial growth and survival can reduce the need for microbiological testing of food, thereby facilitating product formulation and risk assessment at low cost and higher efficiency, ultimately improving the microbiological safety of food [16]. Bakery products including cakes with $a_w$ above 0.82 may support pathogenic growth and survival [17]. Since the cakes are served without heating, consumers may be exposed to bacteria that can easily proliferate upon temperature abuse. However, limited information is available regarding predictive modeling for the outcome of foodborne pathogens in cakes.

Temperature is considered the main factor affecting bacterial growth in food, and temperature control to prevent the growth of bacteria in ready-to-eat (RTE) food, including bakery products, is important. Therefore, this study aimed to develop mathematical models to predict the outcomes of *Salmonella* spp. and *S. aureus* in a cake at varying temperatures, considering that temperature is among the most critical factors influencing bacterial growth in food items during processing, distribution, and storage [18].

## 2. Materials and Methods

### 2.1. Analysis of Water Activity ($a_w$) and Proximate Composition

Nine different types of cake (3 sponge cakes, 3 mousse cakes, 1 cheesecake, 1 brownie, and 1 tiramisu) were purchased from Shinsung Cake & Bakery (Gyeonggi-do, Korea). Cakes used in this study were purchased at the same time. The cake samples were stored at 4 °C until analyses, and the experiment was conducted within 3 days after purchasing. Proximate composition (moisture, fat, protein, and carbohydrate content) and $a_w$ were measured to determine the effect on the growth of *Salmonella* spp. and *S. aureus* in a cake. All cake samples were mixed well prior to analyzing. The $a_w$ of the cakes was determined using a water activity meter (Labmaster-$a_w$ CH-8853, Novasia, Switzerland), and proximate composition was analyzed using standard methods reported by the Association of Official Analytical Chemists (AOAC) [19].

### 2.2. Inocula Preparation

Six strains of *Salmonella* spp. (*Sal*. Enteritidis: ATCC 13076, NCCP 14546, *Sal*. Typhimurium: NCCP 16207, NCCP 12219, *Sal*. Montevideo: NCCP 10140, *Sal*. Kentucky: NCCP 11686) and five strains of *S. aureus* (ATCC 13565 (sea), 14458 (seb), 19095 (sec), 23235 (sed), and KCCM 41324 (see)) were obtained from National Culture Collection for Pathogens (NCCP) and American Type Culture Collection (ATCC). The bacterial strains

were stored at −80 °C and were cultured in 10 mL of tryptic soy broth (TSB; Difco, Becton Dickinson and Company, Sparks, MD, USA) at 35 °C for 24 h. As bacterial growths have obvious strain variation, a mixture of 6 *Salmonella* strains was used as an inoculum, and also a mixture of 5 *S. aureus* strains was prepared. Then, cocktails of *Salmonella* spp. and *S. aureus* strains were serially diluted to 4–5 log CFU/mL, respectively.

### 2.3. Comparisons of Salmonella spp. and S. aureus Growth in Various Cake Samples

Cakes were cut into cubes of $3.5 \times 2.7 \times 2.0$ cm$^3$ (ca., 20 g). Nine different cake samples were placed in a plastic container (HPL847, Lock & Lock®, Asan, Korea), and 0.1 mL aliquots of the inocula were evenly seeded in the cake samples to obtain a final inoculum density of 2–3 log CFU/g. The inoculated samples were stored at 15 °C for 7 d for *S. aureus* or 9 d for *Salmonella* spp. To enumerate *Salmonella* and *S. aureus* cells, each cake sample was transferred to a sterile sample bag (1930FW, 3M, Seoul, Korea) and treated with 50 mL of saline solution (cleancle, JW Pharmaceutical, Dangjin, Korea), followed by pummeling with a pummeler (400 circulator, SEWARD STOMACHER™, Worthing, UK) for 1 min. The samples were then serially diluted (10-fold) with saline solution (3M), and 0.1 mL of the diluent was surface-plated on Xylose Lysine Deoxycholate (XLD, Merck, Darmstadt, Germany) agar and Baird-Parker (BP) agar (Merck) for *Salmonella* spp. and *S. aureus*, respectively. The plates were incubated at 35 °C for 24 h, and the colonies were manually enumerated. Based on bacterial growth, a$_w$, and proximate compositions, cake samples displaying the most rapid growth of *Salmonella* spp. and *S. aureus* were selected for predictive modeling.

### 2.4. Development of Predictive Models for Isothermal Temperature

Cocktails of *Salmonella* spp. and *S. aureus* were inoculated in a selected cake sample as a model. The inoculated samples were stored at 4, 10, 15, 20, 25, and 30 °C for *Salmonella* spp., and 4, 10, 15, 25, and 30 °C for *S. aureus*. The cell counts at each temperature were fitted into the Baranyi model [20] using DMFit 3.5 (Institute of Food Research, Norwich, UK) to determine the maximum specific growth rate ($\mu_{max}$; log CFU/g/h) and lag phage duration (LPD; h). The Baranyi model was as follows:

$$N_t = N_0 + \mu_{max} \times A_t - \ln\left[1 + \frac{\exp(\mu_{max} \times A_t) - 1}{\exp(N_{max} - N_0)}\right] \tag{1}$$

$$A_t = t + \frac{1}{\mu_{max}} \ln\left(\frac{\exp(-\mu_{max} \times t) + h_0}{1 + h_0}\right) \tag{2}$$

where $N_t$ is the cell count (log CFU/g) at any time t, $N_0$ (log CFU/g) is the initial cell count, $N_{max}$ (log CFU/g) is the maximum cell count of *Salmonella* spp. and *S. aureus*, $\mu_{max}$ (log CFU/g/h) is the maximum specific growth rate, $A_t$ is an adjustment function described by Baranyi and Roberts, and $h_0$ is a parameter defining the initial physiological state of the bacteria [20]. Calculated values of $\mu_{max}$, LPD, $N_0$, and $N_{max}$ at each temperature were provided by DMFit based on the Baranyi model. To evaluate the effect of storage temperature on the growth parameters, a polynomial equation was fitted to LPD and $\mu_{max}$ values as a function of temperature. To validate the predictive model developed herein, additional experiments were independently performed at 23 °C, and the observed data were compared with the predicted data obtained from the predictive model developed in this study. Differences between the observed and predicted data were quantified on the basis of the root mean square error (RMSE), bias factor ($B_f$), and accuracy factor ($A_f$), where n is the total number of observations, and *p* is the number of model parameters [21].

$$RMSE = \sqrt{\frac{\sum(\text{predicted value} - \text{observed value})^2}{(n - p)}} \tag{3}$$

$$B_f = 10^{\left[\sum\{\text{Log(predicted values/observed values)}\}/n\right]} \tag{4}$$

$$A_f = 10^{\left[\Sigma\{|\log(\text{predicted values}/\text{observed values})|\}/n\right]} \tag{5}$$

### 2.5. Development of the Dynamic Model

To describe the growth patterns of *Salmonella* spp. and *S. aureus* in cake samples at different temperatures, a time–temperature profile of 10–25 °C was maintained, using a programmable refrigerated incubator (MIR 254, Panasonic Healthcare, Tokyo, Japan), and *Salmonella* spp. and *S. aureus* cell counts in cake were also measured under changing temperature using the same protocol described in 2.3., followed by using as observed data for evaluating model performance. The temperature profile was recorded using a data logger (testo 174T, Testo, Sparta, NJ, USA) and used to predict bacterial growth patterns with the equation reported by Baranyi and Roberts [20]. For dynamic model development, a secondary model for $\mu_{max}$, developed for isothermal temperature, was used for calculating $\mu_{max}$ under changing temperature in the recorded data logger, followed by applying the Baranyi equation to estimate *Salmonella* spp. and *S. aureus* counts in real-time prediction. $h_0$, the value meaning potential growth, was obtained from the average of multiplying $\mu_{max}$ and LPD of the primary model at each temperature. To evaluate the performance of the developed dynamic model, RMSE was calculated. The workflow for the development of isothermal and dynamic models is described in Supplementary Figure S1.

### 2.6. Statistical Analysis

To develop predictive models, all experiments were repeated at least four times, and all data were analyzed using the general linear model procedure of SAS® (version 9.2, SAS Institute Inc., Cary, NC, USA). The differences among fixed effects were determined by pairwise *t*-test at $p = 0.05$.

## 3. Results and Discussion

In nine different cake samples, including sponge cake, mousse cakes, cheesecake, brownie, and tiramisu, which had different $a_w$ and proximate compositions, *Salmonella* spp. adequately survived. Cell numbers of *Salmonella* spp. were maintained at 2–3 log CFU/g for 9 d in all cake samples, except A (sponge cake) and H (brownie) cake (Table 1), where the density of *Salmonella* spp. decreased. In contrast, the growth of *S. aureus* was observed in all cake samples, except for cake A and H. Sample A was marble sponge cake with an $a_w$ of $0.86 \pm 0.00$, which dried more rapidly than other sponge cake samples B and C during storage (data not shown). Sample H showed significantly lower $a_w$ ($0.70 \pm 0.01$) and moisture content ($12.4 \pm 0.1\%$) than other samples. After storage for 7 d, *S. aureus* grew up to 7.5–7.9 log CFU/g in D, E, F (mousse), G (Cheese), and I (tiramisu) cakes, and their $a_w$ values were significantly higher (0.90–0.91) than those of the other samples (Table 1). Growth of *S. aureus* was slightly lower in samples B and C (sponge) than in samples D, E, and F (mousse) because of their low $a_w$ and moisture contents, indicating that *Salmonella* spp. and *S. aureus* growth in cake samples is associated with the $a_w$ and moisture content of the samples. These results suggest that if bakery products, including cake, have a high $a_w$, the growth and survival of *Salmonella* spp. and *S. aureus* are feasible, causing foodborne diseases upon temperature abuse [22]. Cake sample D reflected the worst growth conditions for *Salmonella* spp. and *S. aureus* because they were expected to grow well, considering favorable conditions including $a_w$, moisture content, and bacterial growth. Hence, cake sample D was selected for developing mathematical models. Before bacteria inoculation, total bacteria, *Salmonella* spp. and *S. aureus* cell counts were measured, but no colony was detected in any of the tested cake samples (data not shown).

**Table 1.** Water activity ($a_w$), proximate composition, and bacterial growth/survival in nine different type of cakes.

| Type of Cake | | $a_w$ | Proximate Composition (%) | | | | Bacterial Density (Log CFU/g) [1] | |
|---|---|---|---|---|---|---|---|---|
| | | | Moisture | Fat | Protein | Carbohydrate | *Salmonella* spp. | *S. aureus* |
| Sponge | A | 0.86 ± 0.00 B | 27.7 ± 0.0 F | 16.6 ± 0.4 DE | 8.6 ± 0.1 A | 45.9 ± 0.6 B | 1.5 ± 0.0 C | <LOD[2] D |
| | B | 0.82 ± 0.02 C | 20.4 ± 0.1 G | 31.0 ± 0.6 B | 6.6 ± 0.0 C | 41.3 ± 0.6 C | 2.9 ± 0.1 A | 5.8 ± 0.4 B |
| | C | 0.80 ± 0.02 C | 27.7 ± 0.0 F | 17.2 ± 0.1 D | 7.2 ± 0.0 B | 47.1 ± 0.1 A | 3.1 ± 0.1 A | 5.5 ± 0.4 B |
| Mousse | D | 0.92 ± 0.01 A | 43.9 ± 0.1 B | 14.1 ± 0.4 F | 6.3 ± 0.0 D | 35.0 ± 0.2 E | 3.4 ± 0.9 A | 7.7 ± 0.0 A |
| | E | 0.90 ± 0.01 A | 40.1 ± 0.0 D | 14.2 ± 0.2 F | 5.9 ± 0.0 E | 39.1 ± 0.2 D | 2.6 ± 0.2 AB | 7.5 ± 0.1 A |
| | F | 0.90 ± 0.01 A | 45.1 ± 0.0 A | 10.0 ± 0.1 G | 3.9 ± 0.1 G | 40.6 ± 0.1 C | 2.6 ± 0.2 AB | 7.8 ± 0.1 A |
| Cheese | G | 0.90 ± 0.00 A | 32.1 ± 0.1 E | 23.8 ± 0.6 C | 8.6 ± 0.1 A | 34.7 ± 0.8 E | 2.6 ± 0.2 AB | 7.6 ± 0.1 A |
| Brownie | H | 0.70 ± 0.01 D | 12.4 ± 0.1 H | 34.1 ± 0.1 A | 5.7 ± 0.1 F | 46.4 ± 0.3 AB | 1.9 ± 0.1 BC | 3.5 ± 0.0 C |
| Tiramisu | I | 0.91 ± 0.01 A | 43.3 ± 0.3 C | 16.0 ± 0.5 E | 7.4 ± 0.1 B | 32.3 ± 0.3 F | 3.0 ± 0.7 A | 7.9 ± 0.1 A |

[A–H] Means within the same column with different superscript letters are significantly different ($p < 0.05$); [1] bacterial density in cake after inoculation with *Salmonella* spp. (9 days) and *S. aureus* (7 days); [2] limits of detection (<1.5 Log CFU/g).

To develop a primary model to determine the kinetic parameters ($\mu_{max}$ and LPD), bacterial growth patterns in cake samples were examined at 4, 10, 15, 20, 25, and 30 °C for *Salmonella* spp., and 4, 10, 15, 25, and 30 °C for *S. aureus*, respectively. Only at 20, 25, and 30 °C, *Salmonella* spp. grew in cake samples (data not shown). For *S. aureus*, growth was observed at 15, 25, and 30 °C (data not shown). Growth data observed herein were fitted to the Baranyi model to develop the primary model. The values of $\mu_{max}$ for cake contaminated with *Salmonella* spp. were 0.10 ± 0.00, 0.19 ± 0.00, and 0.35 ± 0.05 log CFU/g/h at 20, 25, and 30 °C, respectively. In the case of *S. aureus*, the values of $\mu_{max}$ were 0.05 ± 0.02, 0.29 ± 0.07, and 0.33 ± 0.02 log CFU/g/h at 15, 25, and 30 °C, respectively. $\mu_{max}$ significantly increased ($p < 0.05$) with an increase in the temperature for both bacteria, indicating that $\mu_{max}$ is temperature-dependent (Table 2). In the case of *S. aureus*, LPD significantly decreased ($p < 0.05$) with an increase in temperature, while no significant differences were observed in the LPD of *Salmonella* spp. above 20 °C, where the growth of *Salmonella* spp. was observed. $R^2$ values of 0.942 to 0.996 indicate that the primary model was appropriate. Secondary models describing the effect of storage temperature on kinetic parameters ($\mu_{max}$ and LPD) were developed, using a polynomial model (Figure 1). For temperatures at which no growth was observed, $\mu_{max}$ was considered zero, and LPD was estimated on the basis of the maximum storage time. The $R^2$ values of the secondary models were 0.968–0.988, indicating that the secondary models were appropriate to describe the effects of temperature on the kinetic parameters. To evaluate the performance of the models developed for *Salmonella* spp. and *S. aureus* in cake samples at a constant temperature, $B_f$, $A_f$, and RMSE were determined. The values of 0 for RMSE indicate perfect concurrence between the predicted data and the observed data, and when the $B_f$ and $A_f$ values were closer to 1, the performance of the developed model was considered a good fit [23]. In this study, $B_f$, $A_f$, and RMSE values of the model for *Salmonella* spp. were 0.98, 1.11, and 0.78, respectively, and those of *S. aureus* were 0.97, 1.07, and 0.54, for *Salmonella* spp., respectively. These validation data indicate that the model developed herein successfully describes the behavior of *Salmonella* spp. and *S. aureus* in cake samples and can be used to develop risk management alternatives to control the risk of foodborne diseases.

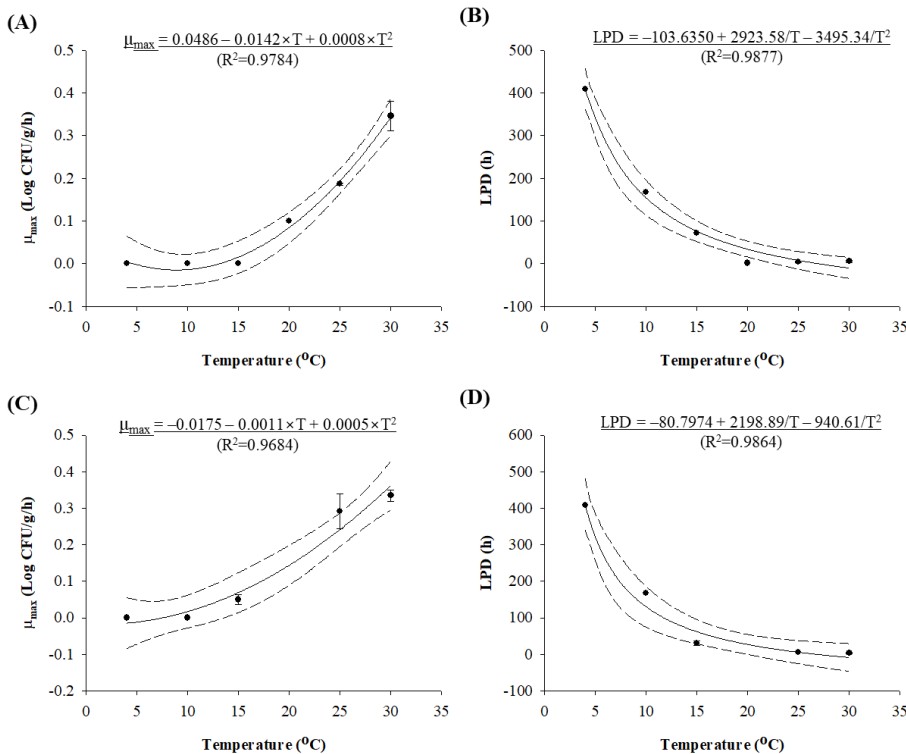

**Figure 1.** Secondary predictive model for the maximum specific growth rate ($\mu_{max}$) and lag phase duration (*LPD*) of *Salmonella* spp. (**A,B**) and *Staphylococcus aureus* (**C,D**) in the cake as a function of temperature.

**Table 2.** Kinetic parameters calculated by the Baranyi model for *Salmonella* spp. and *Staphylococcus aureus* growth in the cake.

| Bacteria | Storage Temperature (°C) | $LPD$ [1] (h) | $\mu_{max}$ [2] (Log CFU/g/h) | $N_0$ [3] (Log CFU/g) | $N_{max}$ [4] (Log CFU/g) | $R^2$ |
|---|---|---|---|---|---|---|
| *Salmonella* spp. | 20 | 1.6 ± 0.0 [A] | 0.10 ± 0.00 [B] | 1.9 ± 0.1 | 7.0 ± 0.0 | 0.990–0.993 |
| | 25 | 4.0 ± 1.9 [A] | 0.19 ± 0.00 [B] | 3.3 ± 0.5 | 7.5 ± 0.0 | 0.992–0.994 |
| | 30 | 5.2 ± 3.8 [A] | 0.35 ± 0.05 [A] | 3.8 ± 0.0 | 7.1 ± 0.3 | 0.955–0.972 |
| *S. aureus* | 15 | 30.1 ± 8.5 [a] | 0.05 ± 0.02 [b] | 3.7 ± 0.0 | 7.2 ± 0.1 | 0.971–0.996 |
| | 25 | 6.3 ± 0.2 [b] | 0.29 ± 0.07 [a] | 4.0 ± 0.3 | 8.1 ± 0.4 | 0.985–0.989 |
| | 30 | 4.2 ± 2.5 [b] | 0.33 ± 0.02 [a] | 3.7 ± 0.7 | 7.6 ± 0.3 | 0.942–0.965 |

[1] Lag phase duration; [2] the maximum specific growth rate; [3] initial cell concentration; [4] maximum cell concentration; [a–b] means within the same column with different superscript letters are significantly different ($p < 0.05$); [A–B] means within the same column with different superscript letters are significantly different ($p < 0.05$).

Sigmoidal growth models are excellent to predict bacterial growth at constant temperatures; however, they have some limitations, e.g., isothermal models are not suitable for varying temperatures. To predict bacterial growth under fluctuating environmental conditions, a dynamic model integrating the primary and the secondary models was developed [20]. Accurate predictions using dynamic models provide valuable industrial data to evaluate their current modules during processing, distribution, and storage, and risk assessment to improve their performance in evaluating the risk of bacterial growth in food products. To describe the kinetic behavior of *Salmonella* spp. and *S. aureus* in cake at varying temperatures, a dynamic model was developed, using calculated kinetic parameters at a constant temperature (Figure 2). In Korea, refrigerated products, including cakes, should be stored and distributed at 0–10 °C [24]. The average temperature in summer in Korea over the last 5 years was 24.5 °C [25]. Therefore, we set the variable temperature range from 10 to 25 °C. During this temperature profile, *S. aureus* cell populations in cake increased from 3.7 to 7.5 log CFU/g for 6 d; however, *Salmonella* spp. did not grow (Figure 2). RMSE

values of the developed dynamic model were 0.22 and 0.19 for *Salmonella* spp. and *S. aureus*, respectively, indicating that the developed dynamic model had a good performance. Therefore, the developed dynamic model can be used to predict *Salmonella* spp. and *S. aureus* growth in preventing foodborne outbreaks.

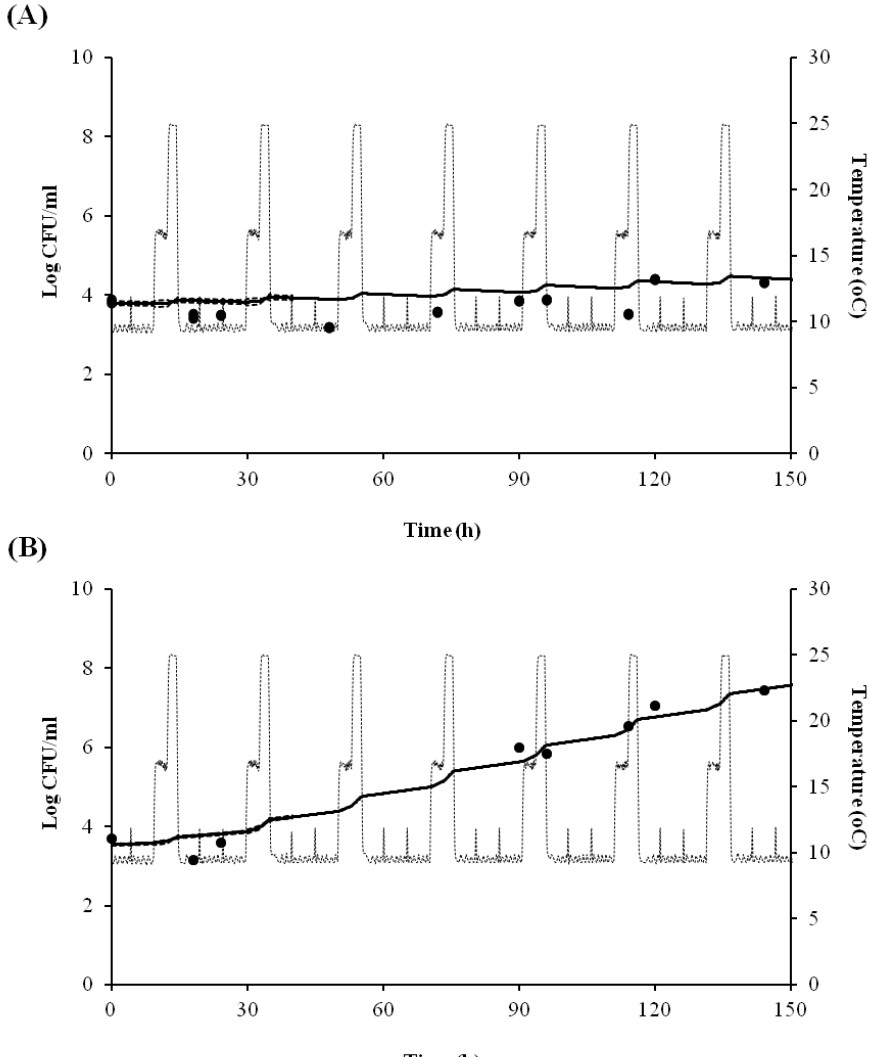

**Figure 2.** Dynamic model for the *Salmonella* spp. (**A**) and *Staphylococcus aureus* (**B**) in the cake during changing temperatures (10–25 °C); symbol: observed value; line: predicted value; dotted line: changing temperatures.

## 4. Conclusions

In conclusion, as expected, the $a_w$ and moisture content of cake are generally associated with the growth of *Salmonella* spp. and *S. aureus*. The models developed in this study can be used to describe the kinetic behavior of *Salmonella* spp. and *S. aureus* in cake at isothermal and changing temperatures and to control the risk of foodborne illness due to cake consumption.

**Supplementary Materials:** The following are available online at https://www.mdpi.com/2076-3417/11/6/2475/s1, Figure S1: The workflow for the development of isothermal and dynamic models.

**Author Contributions:** Conceptualization, H.L. and H.J.K.; methodology, H.L. and J.H.P.; formal analysis, Y.K.P.; writing—original draft preparation, H.L.; writing—review and editing, H.L., J.H.P., and H.J.K.; supervision, H.L. and H.J.K.; project administration, H.L. and H.J.K.; funding acquisition, H.J.K. All authors have read and agreed to the published version of the manuscript.

**Funding:** This research was funded by Korea Food Research Institute, grant number E0210700-01.

**Institutional Review Board Statement:** Not applicable.

**Informed Consent Statement:** Not applicable.

**Acknowledgments:** This research was supported by the Main Research Program (E0210700-01) of the Korea Food Research Institute (KFRI) funded by the Ministry of Science and ICT (Republic of Korea).

**Conflicts of Interest:** The authors declare no conflict of interest.

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
