# Peer review of "Mathematical Modeling for the Growth of Salmonella spp. and Staphylococcus aureus in Cake at Fluctuating Temperatures"

_applsci, doi:10.3390/app11062475_

Round 1
Reviewer 1 Report
In the artcile the models were developed which can be used to describe the kinetic behavior of Salmonella and S. aureus in cake at isothermal and changing temperatures and to control the risk of foodborne illness due to cake consumption. The article presents an interesting issue in the matter of food safety, especially nowadays, where, as we know, we are still threatened by the risk of an epidemic or pandemic. The article is written in a very accessible way, the methodology is described sufficiently, the discussion and discussion of the results as well. After all, I have a few minor comments
In the introduction you devote a lot of space to Salmonella giving different statistical data, maybe it is worth using the same scheme for Sthaplycococcus bacteria?
I think that in the introduction you should also discuss the issue of modeling more widely.
Bacteria names, such as S. aureus and Salmonella, should be in italics - correct throughout the text.
How the strains were stored long term?
Author Response
The authors thank the reviewers for their constructive comments. Our responses to each of the comments are provided in the attached file.

Reviewer 2 Report
The authors examine the growth of two bacteria geni in ready-to-eat cake products. The capacity of growth was tested for various cake types and specific temperature dependent dynamics was examined for a Mousse-type delicacy. With the backdrop of food poisoning events detailed knowledge on food spoilage can improve food safety. The manuscript is well written and coherently unfolds the argument. Some few major revisions are necessary.
- The age of the cakes upon purchase is unknown which could affect the subsequent growth tests. It would be beneficial to examine the starting microbial population before inoculation to assess growth of competing bacteria, or a negative control is missing which could inform of any other growing organisms. Please comment on this.
- The authors should more specifically mention the envisaged implementation of their predictive model. What are the practical implications of their contribution? In what form will it bring maximum benefit? It would be desirable to have this also in the abstract.
- The description of the dynamic model is insufficient. Please provide more detailed descriptions of how the polynomial model for growth rate temperature dependence is combined with the Baranyi Model.
- The experiments for the fluctuating temperature test are improperly described/missing.
- Why does the LPD for Salmonella increase with higher temperature?
- It is important to publish the workflow and the data either as supplementary zip or on a cloud sharing platform (e.g. github, bitbucket, zenodo, etc.)
Minor comments:
L59: write out water activity with correct abbreviation in brackets.
L93,94,110,113: S. aureus in italics.
L117: lag phase duration
L185: 0.29
References:
consider using tiny urls for long urls.
L279: link broken
There are a lot of URLs most of which will break eventually because they have no associated doi.
Author Response

(The authors gave the same response as above.)

Round 2
Reviewer 1 Report
Thank you for addressing the comments.
Reviewer 2 Report
The authors have sufficiently responded to the comments and their manuscript can be published.